# Research on the Influence Mechanism of Intention to Proximity Travel under the COVID-19

**DOI:** 10.3390/bs13010010

**Published:** 2022-12-22

**Authors:** Huan Chen, Luyao Wang, Shaogui Xu, Rob Law, Mu Zhang

**Affiliations:** 1Shenzhen Tourism College, Jinan University, Shenzhen 518053, China; 2School of Management, Jinan University, Guangzhou 510632, China; 3Asia Pacific Academy of Economics and Management, Department of Integrated Resort and Tourism Management, Faculty of Business Administration, University of Macau, Macau 999078, China

**Keywords:** proximity travel, tourists’ risk perception, travel intention, COVID-19, MGB

## Abstract

The outbreak of COVID-19 has brought increasing attention to proximity travel. This mode of travel is a convenient travel setup for both tourists and neighboring destinations. With the help of the model of goal-directed behavior (MGB), this study investigates the influence of tourists’ perception of epidemic risk on their intentions for proximity travel during the normalization of epidemic prevention and control. This study takes Shenzhen, China as the research area, and carried out the investigation in the context of normalization of the epidemic in China. A total of 489 pieces of valid sample data were collected through questionnaire surveys. Statistical analysis software, such as SPSS26.0 and AMOS23.0, were used to analyze the collected data information quantitatively, including descriptive statistical analysis, reliability and validity test, CFA and SEM. The results showed that attitude, subjective norms, positive anticipated emotions, and perceptual behavior control have significant positive effects on travel desire. Travel desire has a significant positive impact on travel intention, whereas negative anticipated emotions have no significant effect on travel desire. Meanwhile, the epidemic risk perception has a significant positive effect on attitudinal travel desire and travel intention. Under the background of the COVID-19, the stronger that the epidemic risk is perceived by tourists, the more the desire and intention to proximity travel are enhanced.

## 1. Introduction

The outbreak of COVID-19 in 2020 triggered a global health crisis. As a result, people have significantly changed their long-term and near-term travel decisions. The uncertainty caused by COVID-19 is so great that it has made travel risk a factor that people have to consider before traveling. Scholars have conducted research on the perceived risk of COVID-19 changing the world [1,2,3] and found that it has caused a shift in decision-making behaviors when traveling [4]. After becoming accustomed to the impact of the epidemic on life, the public gradually has adapted to the existence of the epidemic psychologically and physiologically, thus slowly ignoring the risks of the COVID-19 [5]. One of the most likely outcomes of the crisis is proximity travel, which means traveling around one’s home [6]. Proximity tourism can be interpreted as “local tourism, peripheral tourism,” emphasizing a short distance, closeness to home, and the use of low-carbon travel methods [7,8]. In the traditional traveling hypothesis, driven by novelty seeking, tourists deliberately seek destinations that are a greater distance from their usual residence to achieve the purpose of seeking unique and novel experiences. However, during COVID-19, travel over long distances has been restricted. With increased social and environmental awareness, many potential tourists perceive nearby destinations as “less risky” [9]. Thus, proximity tourism offers an alternative to tourists during the epidemic. COVID-19 has changed the way that people travel. The weakening of global mobility and the increase in local mobility have shifted managers’ focus.

In different situations, tourists will have various behavioral intentions. In particular, in the context of COVID-19, dangerous situations that may cause physical harm will affect tourists’ travel decisions [10]. Therefore, a model should be constructed to extend the current tourist behavior theory [11]. In addition, not many quantitative studies on behavioral intentions during COVID-19 have been conducted. Most of the existing research focused on crisis management, sustainable development, and tourism destination image [12,13,14]. The number of studies related to the topic of proximity tourism in the context of COVID-19 is relatively scarce. Nevertheless, we are convinced that this area of research has potential in discussing sustainable tourism development during the pandemic. COVID-19 has had a significant impact on the global economy. With massive quarantine travel restrictions and increased social distancing, consumer and business spending plummeted, contributing to the global recession [15]. However, the government has made many efforts to reduce the spread of the COVID-19 virus. Public policies must adapt to changes and/or events [16]. Government policies need to adopt appropriate solutions to prevent the spread of COVID-19 and elevate the status of economic activity in tourism. The COVID-19 crisis should be seen as an opportunity to critically rethink the growth trajectory of the tourism industry [17]. This will also have an impact on mass tourism behavior, which will influence the formulation of public policies.

This study attempts to construct a theoretical framework by applying the extended model of goal-directed behavior (EMGB). The study of tourists’ behavioral intentions toward proximity tourism during COVID-19 is explained by considering the variables of tourists’ risk perceptions, attitudes, anticipated emotions, subjective norms, perceived behavioral control, and desire to travel during COVID-19. In addition to responding positively to the call for recovery strategies during tourism-related disasters and crises [18], this study aims to identify the key factors affecting tourists’ intention to travel around their home country during COVID-19 and provides a basis for governments and tourism enterprises to help tourism recover from the crisis.

## 2. Literature Review and Hypotheses Development

### 2.1. Proximity Travel

The widely accepted definition of tourism is to leave one’s usual place of residence for a period of time and engage in leisure-related activities. Proximity travel therefore breaks with this traditional understanding, describing a kind of tourism close to home and re-appreciating one’s own ordinary environment [19]. Since 2010, some scholars have conducted research on proximity travel [20,21]. The concept of proximity travel revolves around a highly popular world where everyone becomes a tourist and every place can become a destination [22], with an emphasis on access to neighboring destinations, excursions, and the use of low-carbon modes of transport [20]. People tend to view distance as a relative dimension, stressing travel experience over physical distance [8,23]. López et al. defined proximity tourism as a tourist experience that occurs in the vicinity of one’s place of residence during the day, so overnight accommodation is not required, with a focus on becoming familiar with unfamiliar places nearby [24]. From the perspective of sustainable tourism, the emergence of proximity travel as a phenomenon is closely related to the increase in environmental awareness [25,26,27] and ushers in new opportunities [8]. Proximity tourists are looking for the closeness of the tourism experience; proximity travel not only promotes the care of proximity tourists to familiar destinations but also strengthens the development of neighboring destinations [7]. The health, social, and economic emergency caused by COVID-19 has provided an opportunity for tour operators to rethink the scope of tourism in a more sustainable and high-quality way, as well as for tourists to rethink the way they enjoy their holidays [28].

Due to limited mobility and social distancing, the tourism industry is particularly vulnerable to pandemic measures [17]. The renewed interest in travel restrictions and closures associated with the COVID-19 outbreak has led to proximity travel gaining increased attention [18]. In the context of COVID-19, Rantala et al. suggested a newer way to understand tourism [20]. Many scholars believe that one of the consequences of the COVID-19 crisis has been the promotion of proximity travel, which has also limited travel to one day or a week [9,29]. As the public adjusts to COVID-19, travelers are opting for closer destinations when long-distance travel is no longer feasible [7]. Home-based tourists can effectively improve their psychological capital in a short period through this short-distance leisure method [30]. This mode is a necessary way to recover physically and mentally during COVID-19. Moreover, as proximity travel is not as susceptible to globalization as international tourism, it can be used as an alternative strategy to international tourism [31]. It also helps to solve some problems in tourism practices, such as seasonality [32] and negative environmental impacts [20]. In the context of the epidemic, combined risk research has been completed by many scholars. Rahman et al. found that travel risk and management perception is significantly associated with risk management, service delivery, transportation mode, distribution channels, destinations to avoid overpopulation, and health and safety, and identified the mediating role of travel risk and management cognition [33]. COVID-19 has had a much more devastating impact on the travel industry. Tourism managers must carefully assess the impact of the pandemic on businesses and develop new risk management methods to respond to the crisis [16].

### 2.2. The Model of Goal-Directed Behavior

The model of goal-directed behavior (MGB), developed from the theory of planned behavior (TPB), was first used to examine content related to psychology. MGB was developed by Perugini and Bagazzi [11]. Emotions, motivational processes, and past behaviors were incorporated into TPB, anticipated emotions were added as the pre-variable, and behavioral desires were set as the mediating variable. Among them, the expected sentiment variables are divided into positive anticipated emotion (PAE) and negative anticipated emotion (NAE) [34]. Behavioral desire (DE) is an individual’s mental state, which is the individual’s expectation that specific events and behaviors will occur [35]. After adding behavioral desire variables, it can play a better integration role in attitude, subjective norms, and perceived behavioral control to predict individual behavior more effectively [11]. Thus, a relatively complete goal-oriented model is roughly formed, as shown in Figure 1.

In the field of tourism research, the extended model of goal-directed behavior (EMGB) is widely used to study the decision-making process of travelers [36,37]. EMGB can predict the behavior of tourists during conditions of risk and uncertainty [38,39,40]. On the basis of the background of TPB and EMGB, this study takes tourists’ behavioral intention to proximity travel as the final result variable. Considering the influence of actual conditions on past behavior frequency variables during COVID-19, Lee et al. used the goal-oriented model to study the decision-making process of Korean tourists’ wine tourism and found that the frequency of past behavior had no significant impact on tourism intention [41]. This study deleted the past behavior frequency variable based on the MGB. When applying the MGB, the model can be supplemented by adding some external variables that can directly or indirectly affect the individual’s behavioral intention to give full play to the actual effect of the model application [42]. For example, Kim et al. added risk perception variables to the goal-oriented model when studying the impact of perceived risk during the Hong Kong protests on the travel decisions of international tourists, explaining the reasons why tourists did not travel to Hong Kong [40]. This study draws on the conceptual model by Kim et al. [40] to increase the risk perception (RP) variable and takes attitude (AT) as the mediating variable of RP and DE. Meanwhile, DE is the mediating variable of RP and TI. The conceptual model of this study is shown in Figure 2.

### 2.3. Hypotheses Development

#### 2.3.1. Risk Perception, Attitude, Travel Desire, and Travel Intention

Risk perception is the core problem that affects tourists’ behavior and decision-making [43]. It refers to the potential loss determined by the individual, including the likelihood of causing each outcome [44]. Horne pointed out that RP is an important predictor of travelers’ behavior [45]. It has a significant impact on the overall image of the destination and the behavioral intent of tourists [46]. Wang et al. focused on tide observation in China to study risk perception [47]. Their research applied and extended Rimal and Real’s attitudinal framework for risk perception [48]. Kong found that the four main risks perceived by millennial Chinese women when traveling were sex/race, betrayal of nationalism, loss of face, and parent–child conflict [49]. Khan constructed a comprehensive model of future young women’s travel behavior based on cognitive and emotional perceptions of destinations, travel motivations, perceived risks, and travel restrictions [50]. Lee’s research suggested that tourists’ perceptions of H1N1 2009 did not limit potential tourists’ desire for international travel because they had some adaptive behaviors that reduced the threat of infection to their acceptable level [51]. Cheng and Yin constructed an S-O-R model using RP as a mediating variable to explore the intention of tourists in wellness tourism [52]. Li et al. further understood the potential behavioral shifts of tourists by reviewing psychological distance and explanatory level theories, as well as the relationship between psychological distance and perceived risk [53]. In this study, RP refers to people’s perceived travel risk in the context of COVID-19.

Attitude (AT) is regarded as an evaluation dimension, representing the evaluation of the entity under review [54]. AT toward a behavior is most favorable when people believe that it will lead to beneficial and pleasant results [44]. Under normal epidemic prevention and control, people feel physical and mental pressure due to home isolation and worries about COVID-19. Thus, they urgently need a way to release the pressure. Proximity travel can meet this demand. This kind of tourism activity is positive, meaningful, and valuable in the eyes of tourists. Based on the above discussion, the following hypothesis is proposed.

**H1.** *The risk perception of COVID-19 positively influences tourists’ attitude toward proximity travel*.

In the MGB, travel desire (DE) is defined as the key factor that explains the formation of a person’s decision [55]. Taylor added the variable of expected regret (RP) to the MGB under conditions of uncertainty and risk, demonstrating its significant effect on the DE [38]. Kim found that travelers’ perceived travel risks during the Hong Kong protests positively influenced the desire not to travel to Hong Kong during this period [40]. Therefore, a significant relationship between RP and DE can be demonstrated. Under the control of COVID-19, the stronger the risk of COVID-19 that is perceived by tourists, the greater the tourists’ depression will be, and the more eager tourists will be to go out. As the fastest way to achieve tourism, proximity travel will be the first choice for tourists. On this basis, this study proposes the following hypothesis.

**H2.** *The risk perception of COVID-19 positively influences tourists’ desire to proximity travel*.

Behavioral intention is considered an agentic behavior as long as a high correlation exists between intention and behavior. We can predict an individual’s behavior from his or her attitude [54]. According to the existing research, travel intention (TI) is affected by many factors, such as product image, national image, and ethnocentric hostility [56]; immigrant animosity [57]; and subjective knowledge [58]. Previous studies have shown that the RP associated with a disease crisis can greatly affect the travel behavior and intention of tourists during the crisis [53,59]. Given the convenience of the form of proximity travel, it is popular among tourists during the long-term fight against COVID-19. Therefore, the following hypothesis can be proposed.

**H3.** *The risk perception of COVID-19 positively influences tourists’ intention to proximity travel*.

#### 2.3.2. Attitude, Subjective Norm, Anticipated Emotion, Perceived Behavior Control, and Travel Desire

Before an individual performs a certain behavior, he or she tends to pre-judge the benefit or loss that this behavior will bring. This pre-judgment creates the individual’s attitude, which reflects the psychological desire to perform the behavior [55]. According to the MGB, AT has an indirect positive effect on behavioral intention through behavioral desire. In the field of tourism research, a number of studies have confirmed that AT has a positive impact on DE. Lee et al. found in the decision-making process of tourists in heritage tourism destinations that AT had a positive effect on DE [60]. In addition, Meng and Choi found that AT had a positive effect on DE in slow travel [30]. Meanwhile, Hang found that tourists’ AT toward the pro-environmental decision-making process in cruise tourism had a positive effect on DE [61]. On the basis of the above research, this study proposes the following hypothesis.

**H4.** *AT has a positive influence on tourists’ desire to proximity travel*.

Subjective norms (SN) are defined as the perceived characteristic pressures of performing or not performing a particular behavior [62]. The source of this stress is the opinion and attitude of family and friends [63]. Previous studies have proven that SN have a significant impact on DE. Kim et al. confirmed that tourists’ SN positively influenced their behavioral desires [40]. Song et al. found that tourists’ SN about eco-friendly festivals had a positive impact on their behavioral desire to revisit the festival [55]. Meng and Choic showed that tourists’ SN can significantly influence their behavioral desire to travel by bike [36]. When studying gaming tourism, Ji found that the SN of Chinese tourists’ gaming consumption behavior positively affected their behavioral desires [64]. Based on the above research, this study proposes the following hypothesis.

**H5.** *SN has a positive influence on tourists’ desire to proximity travel*.

The variable of individual emotion, anticipated emotion, is added into the MGB. That is, when facing a certain behavior, an individual will consider the changes that his or her behavior will bring to his or her emotion [11]. Anticipatory emotion is divided into positive anticipatory emotion (PAE) and negative anticipatory emotion (NAE) [34]. In this study, PAE refers to the positive emotion generated by tourists participating in proximity tourism, while NAE refers to the negative emotion generated by tourists not being able to participate in proximity tourism. In previous studies, travel-related PAE and NAE significantly influenced DE [65,66]. The following hypotheses are then proposed.

**H6.** *PAE has a positive influence on tourists’ desire to proximity travel*.

**H7.** *NAE has a positive influence on tourists’ desire to proximity travel*.

Perceived behavioral control (PBC) is the confidence or ability of an individual to perform a certain behavior, which is considered the necessary factor of behavioral intention and actual behavior [55]. When an individual judges that he or she has sufficient conditions to perform a certain behavior, his or her desire to perform the behavior will be enhanced [67]. Yuan and Yoo showed that PBC positively influenced people’s desire to use Airbnb [31]. In addition, during conditions of uncertainty and risk, Taylor confirmed that visitors’ PBC affected how eager they were to search for information [38]. Therefore, this study proposes the following hypothesis.

**H8.** *PBC has a positive influence on tourists’ desire to proximity travel*.

#### 2.3.3. Travel Desire and Travel Intention

Tourists’ behavioral intention (TI) has been found to be related to tourists’ travel desire. Desire has been proven to be an important predictor of tourists’ future travel behavior intention [68]. Huang et al. found a significant positive effect of desire to travel on intention to travel in an empirical study of tourists who had travel experiences after experiencing a public health emergency [69]. The following hypothesis is proposed in this study. AT often acts as an intermediary variable to influence the dependent variable. Han and Al-Ansi examined the mediating role of AT between SN and TI [59]. Fu et al. tested the mediating role of attitude as a mediating variable of SN and praise behavior in online shopping [63]. In addition, Kim et al. constructed an MGB to examine the indirect influence of perceptual morality on behavioral desire through AT [65]. On the basis of the above analysis, this study proposes the following hypothesis. DE is often used as an intermediary variable to influence TI. Kim et al. used DE as a mediating variable to verify the perceived risk of tourists’ willingness to travel to Hong Kong [40]. Han et al. set DE as a mediating variable between past behavior frequency and TI [55]. Furthermore, Song et al. found that DE played a mediating role in the influence of PBC on TI [55]. Therefore, the following hypotheses are proposed in this study.

**H9.** *DE has a positive influence on behavioral intention of proximity travel*.

**H10.** *AT mediates between the risk perception of COVID-19 and travel desire*.

**H11.** *DE mediates between the risk perception of COVID-19 and TI*.

## 3. Methodology

### 3.1. Research Environment

Since the outbreak of COVID-19 in early 2020, countries have adopted different strategies to respond [70]. China’s tourism position is going through a difficult recovery process as the epidemic situation evolves. On 20 January 2020, the coronavirus was found to be a risk of human-to-human transmission. “Quarantine” became the key word worldwide, which brought the development of tourism to a standstill. On 8 April 2020, the lifting of the lockdown in Wuhan, China, marked a new stage in epidemic prevention and control. However, safety concerns remain the most important issue for people to travel. On 14 July 2020, China gradually resumed inter-provincial group travel, as well as air tickets and hotels. As the epidemic gradually improved, tourism recovered, but the emergence of many epidemic cases in China made recovery difficult. Under the guidance of the continuous proactive policy, China downgraded the prevention and control situation from emergency management in early 2020 to routine prevention and control. Chinese citizens are gradually returning to normal life. This study is carried out research in the context of “normalized epidemic prevention and control” in China.

### 3.2. Measurement

The first part of the questionnaire explains the proximity travel, that is, daily local travel and peripheral travel refer to a short and low-carbon way of travel. The second part contains questions for all variables. The third part contains demographic information about the survey respondents.

As the research was conducted in China, the questionnaire followed the standard procedure of “translation–back translation” [71]. All measurement questions were translated into Chinese. Experts proficient in tourism research and English were invited for a group discussion to ensure the content validity of the questionnaire. All the scales in the questionnaire used a Likert five-point scale, ranging from 1 (strongly disagree) to 5 (strongly agree).

In the second part of the questionnaire, for the RP scale, Cho et al. measured the RP from the perspective of perceived vulnerability and perceived severity when comparing the impact of psychological factors, such as the public risk perception on protective behavior in the United States and South Korea [72]. Seale et al. studied the Australian public’s perception of H1N1 risk in 2009 as measured by four questions on risk severity [73]. On the basis of the background of COVID-19, the structure of the RP scale followed the public risk perception scale developed by Dai et al. for public health emergencies [74]. Ten items were measured in terms of outbreak severity, controllability, severity, and likelihood of health impacts. The MGB scale had 26 entries in 7 dimensions. For the scale with 3 dimensions of SN, AT, and PBC, this study mainly referred to the study of Han et al. [61], including 12 items. PAE, NAE, and DE referred to the research of Lee et al. and Song et al. [50,55], with a total of 10 entries. TI referred to the study by Kim et al., which included four entries [40].

The self-reporting questionnaire used in this study might be affected by common method variance (CMV) and might even lead to erroneous conclusions [75,76]. Therefore, this study attempted to reduce the influence of CMV through some program control. First, this study collected questionnaires through random sampling and filled in questions anonymously to reduce CMV at the source to protect privacy. Second, in the process of designing the questionnaire, this study placed the independent variable first and the dependent variable later [77]. Third, Harman’s single-factor test was used to evaluate CMV, and the results showed that the first factor accounted for 26.67% of the total variance. If it was less than 40% of the total variance, then we could consider that no serious common method bias occurred [78,79]. Therefore, the CMV in this study did not affect the results of the quantitative study.

### 3.3. Sample and Data Collection

In this study, two academic experts, two doctoral students, and three master’s students were invited to form focus groups to evaluate the applicability of the original questionnaire. On the basis of the focus group feedback, minor changes were made to the description and wording of the questionnaire, as well as adjustments to the layout of the questions, thus resulting in the formal questionnaire. To check the level of understanding and appropriateness of the structure of the questionnaire, 80 questionnaires were distributed online for pre-survey on 26 February 2021. A total of 76 valid questionnaires were recovered, with an effective recovery rate of 95%. The reliability and validity test of the pre-survey questionnaire data showed that the Cronbach’s α values were greater than 0.8 and the factor load was greater than 0.7 in all dimensions. Thus, the scale ensured good internal consistency. Preliminary survey results indicated that the scale was reliable and effective [80,81].

The official research period of this study was from 19 April to 7 May 2021. Given the requirements of epidemic prevention and control during the investigation period, the number of personnel had to be limited. Therefore, this study adopted the convenient snowball sampling design to issue questionnaires with the help of the questionnaire survey platform. The online questionnaires are distributed by Wen Juan Xing (https://www.wjx.cn/ (accessed on 8 May 2021)). At the same time, the researchers forwarded and promoted it on multiple social platforms such as WeChat, and obtained effective data on potential tourists. As for the sample size of the questionnaire, Gorsuch believed that the ratio of the number of questionnaire items to the sample size was 1:5 [82]. Yu et al. believed that valid samples should be greater than 100 [83]. A total of 36 questions were included in this study. In view of the cost of time, manpower, and economy in the study, 550 samples were actually collected. After excluding 61 invalid questionnaires, a total of 489 valid questionnaires were obtained, with an effective rate of 88.9%.

## 4. Data Analysis

### 4.1. Demographic Analysis

First, SPSS22.0 was used to perform descriptive statistical analysis on 489 data samples. In the effective samples, 42.1% of men and 57.9% of women were included, and the gender distribution was basically equal. The age group of respondents was mainly distributed between 18 and 50 years old, and most of them were concentrated between 18 and 25 years old. The majority of respondents were students (35.8%) or worked in enterprises (33.5%). In addition, most had a bachelor’s degree or above (64.2%), thus indicating that the respondents had a good level of education.

### 4.2. Reliability and Validity Analysis

Confirmatory factor analysis (CFA) was performed to evaluate the effectiveness of the model. SPSS26.0 and AMOS23.0 were used to process the data to evaluate the reliability and validity of the model. The commonly used reliability test method is the Cronbach’ α reliability coefficient method. If the Cronbach’ α values are greater than 0.8, the data are reliable. In this study, the overall number of questions on the scale was 36, and the Cronbach’ α value was 0.899, thus indicating that the reliability of the data was high. The Cronbach’s α values of nine dimensions of the questionnaire, namely RP, AT, SN, PBC, PAE, NAE, DE, and TI, were tested, and the reliability coefficients of each dimension were above 0.7. The internal consistency of the questionnaire items was high.

The convergence validity is a test of the relationship between measurement items within the same variable, mainly to examine the degree of correlation between the items [84]. Studies have shown that convergence validity mainly measures two indicators, mean variance variation (AVE) and combinatorial reliability (CR) [85]. According to the research, the AVE value should be at least greater than 0.5. The higher the AVE value is, the more explanatory power of each item inside each variable is for that variable. CR refers to the consistency between the items within each variable, which should be at least greater than 0.6. As can be seen from Table 1, the factor load of each item was greater than 0.7, thus indicating that the topic was representative. At the same time, the AVE of each latent variable was above 0.5 and CR above 0.7, thereby explaining that the convergence validity was ideal.

As for the discriminant validity, according to the findings of Fornell and Larcker, to test the discriminant validity of a variable, the mean variance variation of the variable and the absolute value of the correlation coefficient with other variables must be compared. When the former is greater than the latter, the variable has good discriminating validity [86]. According to the data (Table 2), the correlation coefficient was smaller than the square root of AVE. Thus, a certain correlation and discrimination existed between the latent variables, and the discrimination validity was ideal.

### 4.3. Hypothesis Test

#### 4.3.1. Path Analysis

The model was analyzed using AMOS23.0 and showed a generally good fit to the data (χ^2^/df = 1.87 < 3, RMSEA = 0.042 < 0.08, CFI = 0.952 > 0.90, TLI = 0.947 > 0.90, IFI = 0.952 > 0.90, GFI = 0.904 > 0.80), as shown in Table 3. In this study, the maximum likelihood estimation method was used to estimate the free parameters of the structural equation model to obtain the path coefficient of tourists participating in the proximity travel. First, the relationship between each variable and its corresponding question item was analyzed. The path coefficient *p* value between each question item and the corresponding question item was significant, and the measurement of each question item on the corresponding variable was effective. The results of the path analysis are shown in Table 4.

Table 4 demonstrates that the standardization coefficient of RP and AT was 0.618, *p* < 0.005, thus indicating that the RP of proximity travel tourists had a significant positive impact on AT. Therefore, H1 was accepted. The standardization coefficient of RP and DE was 0.193, *p* < 0.01, thus indicating that the RP of tourists had a significant positive impact on DE. Therefore, H2 was accepted. The standardization coefficient of RP and TI was 0.625, *p* < 0.001, thus indicating that RP had a significant positive impact on TI. Therefore, H3 was accepted. The standardization coefficient of AT and DE was 0.171, *p* < 0.005, thus indicating that the AT of tourists had a significant positive effect on DE. Hence, H4 was supported. The standardization coefficient of SN and DE was 0.144, *p* < 0.005, thus indicating that the SN of tourists had a significant positive effect on DE, and H5 was supported. The standardized coefficients of PAE and DE were 0.224, *p* < 0.005, thus indicating that the PAE of tourists had a significant positive effect on DE. Therefore, H6 was accepted. The normalization coefficient of NAE and DE was −0.053, *p* > 0.05. Hence, H7 was rejected. The standardization coefficient of PBC and DE was 0.201, *p* < 0.005, thus indicating that the PBC of tourists had a significant positive effect on DE. Hence, H8 was supported. The standardization coefficient of DE and TI was 0.179, *p* < 0.005, thus indicating that the TD of tourists had a significant positive effect on TI. Therefore, H9 was accepted.

#### 4.3.2. Mediating Effect Analysis

Four common test methods are used for the mediation effect. In essence, these four methods are the same. However, they differ in the correction methods of confidence degree. In this study, the optimal bootstrap method was used to test the mediating effect. Sampling was repeated 5000 times by bootstrap to achieve more stable statistical results. Then, the confidence interval in the results was judged. If the interval value of the upper and lower limits of the confidence interval did not contain 0, the mediating effect of this path was significant.

In this study, two mediating effects were examined: RP → AT → DE, RP → DE → TI, and the analysis results are shown in Table 5 below. The RP of COVID-19 affected DE through AT. The indirect effect was a × b = 0.151 (*p* < 0.05), and the confidence interval was [0.169, 0.358], excluding 0, thus indicating a significant mediating effect. Therefore, H10 was accepted. The indirect effect a × b = 0.105 (*p* < 0.05), the confidence interval was [0.405, 0.585], excluding 0, indicating a significant mediating effect, that is, DE had a mediating role between RP and TI, and H11 was established.

#### 4.3.3. Indirect Effect and Total Effect Analysis

This study had several mediating variables, and the relative size of specific mediating effects could be determined. As shown in Table 6, the indirect mediating effect of attitude between RP and DE a × b = 0.151, the total effect of RP and DE was 0.414, and the indirect effect of AT accounted for 0.151/0.414 × 100% = 36.4%. That is, in the indirect influence relationship between RP and DE, 36.4% of the indirect effects were mediated by AT. The indirect mediating effect of DE between RP and TI a × b = 0.105, the total effect of RP and TI was 0.601, and the total effect of DE accounted for 0.105/0.601 × 100% = 17.5%. That is, in the indirect influence relationship between RP and TI, 17.5% of the indirect effects were caused by the mediation of DE. The indirect mediating effect of DE between SN and TI a × b = 0.052, the total effect of SN and TI was 0.352, and the total effect of DE accounted for 0.052/0.352 × 100% = 25.1%. That is, in the indirect influence relationship between SN and TI, 25.1% of the indirect effects were caused by the intermediary effect of DE. The indirect mediating effect of DE between PAE and TI a × b = 0.077, the total effect of PAE and TI was 0.403, and the indirect effect of DE accounted for 0.077/0.403 × 100% = 27.7%. That is, in the indirect influence relationship between PAE and TI, 27.7% of the indirect effects were mediated by DE. The indirect mediating effect of DE between PBC and TI a × b = 0.068, the total effect of PBC and TI was 0.348, and the indirect effect of DE accounted for 0.068/0.348 × 100% = 29.4%. That is, in the indirect influence relationship between PBC and TI, 29.4% of the indirect effects were caused by the mediation of DE.

## 5. Discussion and Implications

### 5.1. Discussion

At present, the research on the influencing factors of tourism mainly focuses on the individual needs of tourists [87], motivation [88], and the push and pull factors of tourism [89,90]. Some research supports the promotion of domestic tourism during COVID-19 [91]. This study takes the EMGB as the theoretical basis, which helps to explore the use of RP as an antecedent variable to investigate the decision-making process of tourists, focusing on the change of tourists’ intention to the proximity travel under the influence of COVID-19. SN, PAE, and PBC have been found to have a significant positive impact on DE. DE has a significant positive impact on TI, whereas NAE has no significant impact on DE. In addition, RP has a significant positive impact on AT, DE, and TI.

RP is an important predictor of TI during COVID-19 [92]. H1, H2, and H3 are accepted, indicating that RP has an effect on AT, DE, and TI, which is consistent with the obtained results [93,94]. The greater the perceived risk is of an activity, the more likely people are to avoid it. This avoidance is not only human nature and instinctive behavior but also a common choice under the background of health and safety threats. H4, H5, H6, and H8 are accepted, showing that AT, SN, PBC, and PAE have a significant impact on DE, which confirms the earlier research [36,51,95,96]. During the epidemic period, the higher the tourists’ evaluation of proximity travel is, such as the view that proximity travel is positive, valuable, attractive, and pleasant [96], the stronger the tourists’ desire to travel will be. When tourists expect that participating in proximity travel can bring them pleasant emotional experiences, such as happiness, joy, excitement, etc., their desire to participate in proximity travel will also increase. The current slow recovery in tourism is mainly due to the instability of COVID-19. Individuals are more likely to trust the suggestions of relatives and friends. Their behaviors will also be subtly influenced. Therefore, individual subjective norms affect the desire to travel. In terms of PBC, when individuals cannot be affected by external factors and control the autonomy of their own lives, they can achieve their wishes and goals according to their own abilities. Under the background of COVID-19, tourists’ desire to participate in tourism activities increases. The research results of the impact of PAE on DE have similarities with the research of Wang et al. [96]. That is, positive expectation is a positive forward-dependent variable of travel intention. However, H7 is rejected, so that NAE did not significantly affect DE to proximity travel, which is different from some conclusions of previous studies. The strong desire of tourists to participate in tourism activities during the current pandemic conditions is related to the fact that being close to nature is a desire that can overwhelm other ideas. H9, H10, and H11 are accepted, so this study verifies the mediating role of AT and DE in tourists’ proximity travel behaviors. Only by meeting the needs of tourists while further promoting the formation of tourism desire can more potential tourists be attracted. This notion has been considered in many scholars’ push and pull factors for tourism, that is, identifying the main motivation driving the desire to travel and attracting travelers to a specific destination can improve the intention to travel [90].

### 5.2. Theoretical Contributions

First, this study incorporates tourists’ intention to proximity travel during COVID-19 into MGB to address the issues on how the AT, SN, PAE, and PBC in epidemic risk perception and MGB affect tourists’ intention to participate in proximity travel. This model extension provides further research on RP and TI. Second, this study expands on the current growing literature on COVID-19 by investigating the impact of tourists’ intention to proximity travel during China’s pandemic policy. COVID-19 has had the most significant impact on travel destinations, such as travel industry cancellations, facility closures, air and cruise route cancellations, and lack of access to tourist destinations [97,98,99]. Although many relevant studies have examined the impact of the epidemic on tourism, research on specific tourism behaviors is still very limited, especially the proximity travel research, which has attracted much attention in the context of the epidemic. In view of the current epidemic situation and the development of tourism, tourists’ proximity travel behavior must be tested from the theoretical level. Third, this study enriches the research on tourists’ proximity travel. At present, the research on recreational belts around cities mainly focuses on the cities themselves, exploring the distribution status of recreational belts around cities, the space-time evolution, and the formation mechanism. Researchers seldom pay attention to the behavior of tourists in close-range tourism activities. During the period of normal epidemic prevention and control, the travel scope of tourists is limited, and proximity travel has become the preferred way for tourists to travel. This study analyzes the influencing factors of the behavior of tourists from the perspective of tourists’ intention to proximity travel and adds relevant research to this field.

### 5.3. Practical Implications

COVID-19 has changed the way people travel, and the reduction in global mobility and the increase in local mobility have shifted managers’ focus. On the one hand, from the perspective of tourism destination management, better control of the epidemic is the prerequisite for better development of tourism. More tourists can only be attracted when the epidemic is well controlled and the risk perception of tourists is reduced. At the same time, on the basis of epidemic prevention and control, it is possible to create a positive image, enhance the attractiveness of proximity travel destinations, and promote it. When executed in this way, people’s positive expectations can be stimulated, the transformation of positive emotions into tourism desire and intention to travel can be promoted, and the vigorous development of proximity travel can be realized. On the other hand, tourists prefer personalized travel methods and leisure items and pay attention to the quality of travel in the post-epidemic period. Therefore, in terms of product content, new locations and new gameplay can allow city tours to demonstrate a different style. In the period of normalization of epidemic prevention and control, tourists can explore new ways to travel around the city in the release of tourism demand. The tourism consumption affected by the epidemic is exhibiting a strong rebound force, which has also brought new sparks to the tourism market. In the period of the normalization of epidemic prevention and control, tourists can explore new ways to travel around the city in the release of tourism demand. The tourism consumption affected by the epidemic is demonstrating a strong rebound force, which has also brought new sparks to the tourism market.

### 5.4. Limitations and Future Research

Some limitations in this study need to be addressed in future studies. First, this work only analyzed the sample data of Shenzhen, China. Therefore, the regional universality of the research conclusions is still insufficient. In future research, the research area can be expanded to select tourism destinations of different types and levels of development in different regions for investigation. Researchers should also consider expanding the research framework in a cross-cultural context, testing it in a more diverse sample pool to enhance the external validity of the findings. Second, the research data were one-time cross-sectional data collected through a self-reported format with certain bias interference. Future research should consider other research designs, such as longitudinal or experimental studies, to address this limitation. Research on other aspects of new forms of tourism around cities that emerged during the normalization of epidemic prevention and control should be explored in the future.

## Figures and Tables

**Figure 1 behavsci-13-00010-f001:**
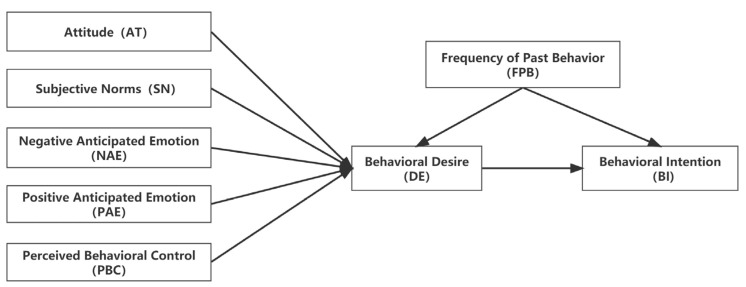
Goal-oriented behavioral theory model.

**Figure 2 behavsci-13-00010-f002:**
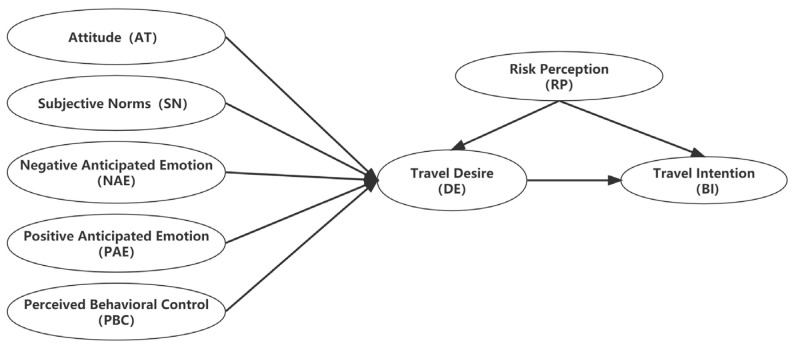
Research model.

**Table 1 behavsci-13-00010-t001:** Confirmatory factor analysis.

Variable	Items	Mean	Factor Loading	Cronbach’s α	CR	AVE
RP	SE1	3.82	0.82	0.729	0.85	0.65
	SE2	3.80	0.80			
	SE3	3.98	0.79			
	CO1	3.78	0.83	0.734	0.86	0.67
	CO2	3.68	0.79			
	CO3	3.74	0.84			
	SHE1	3.69	0.80	0.814	0.82	0.69
	SHE2	3.68	0.86			
	PA1	3.74	0.87	0.792	0.79	0.66
	PA2	3.61	0.75			
AT	AT1	3.91	0.77	0.863	0.86	0.61
	AT2	3.64	0.80			
	AT3	3.78	0.77			
	AT4	3.57	0.79			
SN	SN1	3.49	0.80	0.875	0.88	0.64
	SN2	3.58	0.80			
	SN3	3.55	0.80			
	SN4	3.47	0.80			
PBC	PBC1	3.83	0.81	0.839	0.84	0.64
	PBC2	3.75	0.79			
	PBC3	3.40	0.80			
PAE	PAE1	3.73	0.83	0.877	0.88	0.64
	PAE2	3.64	0.79			
	PAE3	3.62	0.80			
	PAE4	3.64	0.79			
NAE	NAE1	3.02	0.79	0.879	0.88	0.65
	NAE2	2.96	0.81			
	NAE3	2.91	0.81			
	NAE4	3.08	0.81			
DE	DE1	3.86	0.77	0.815	0.82	0.60
	DE2	3.51	0.81			
	DE3	3.67	0.74			
TI	TI1	3.89	0.80	0.879	0.88	0.65
	TI2	3.59	0.80			
	TI3	3.58	0.81			
	TI4	3.71	0.82			

**Table 2 behavsci-13-00010-t002:** Tests of discriminant validity.

	SE	CO	SHE	PA	SN	PBC	PAE	NAE	AT	DE	TI
SE	0.81										
CO	0.50 **	0.82									
SHE	0.47 **	0.44 **	0.83								
PA	0.48 **	0.48 **	0.41 **	0.81							
SN	0.15 **	0.17 **	0.15 **	0.15 **	0.80						
PBC	0.23 **	0.18 **	0.19 **	0.19 **	0.19 **	0.80					
PAE	0.20 **	0.17 **	0.20 **	0.27 **	0.19 **	0.26 **	0.80				
NAE	0.05 *	0.06 *	0.08 *	0.03 *	0.00 *	0.12 *	0.02 *	0.81			
AT	0.32 **	0.34 **	0.32 **	0.34 **	0.35 **	0.37 **	0.41 **	0.09 *	0.78		
DE	0.27 **	0.31 **	0.29 **	0.30 **	0.30 **	0.35 **	0.38 **	0.01 *	0.43 **	0.77	
TI	0.47 **	0.42 **	0.35 **	0.40 **	0.38 **	0.38 **	0.42 **	0.04 *	0.45 **	0.41 **	0.81
AVE	0.65 **	0.67 **	0.69 **	0.66 **	0.64 **	0.64 **	0.64 **	0.65 **	0.61 **	0.60 **	0.65 **

Note: The values on the diagonal line are the square roots of AVE, whereas those off the diagonal line are the inter-construct correlation coefficients. * *p* < 0.05, ** *p* < 0.01.

**Table 3 behavsci-13-00010-t003:** Results of assessing alternative models.

	Χ^2^/df	RMSEA	TLI	CFI	AGFI	GFI
Values	1.87	0.042	0.947	0.952	0.887	0.904
Optimal value	<3.0	<0.08	>0.90	>0.90	>0.80	>0.80
Standard value	3.0~5.0	0.08~0.10	0.70~0.90	0.70~0.90	0.70~0.90	0.70~0.90

Notes: χ^2^/df, goodness-of-fit test; RMSEA, root mean square error of approximation; TLI, Tucker–Lewis; CFI, comparative fit index; GFI, goodness-of-fit index; AGFI, adjusted goodness-of-fit index.

**Table 4 behavsci-13-00010-t004:** Path analysis.

Hypothesis	Estimate	Standard Error	CR	*p*	Result
H1	0.618	0.069	9.795	***	Accepted
H2	0.193	0.087	2.345	*	Accepted
H3	0.625	0.079	9.014	***	Accepted
H4	0.171	0.063	2.623	**	Accepted
H5	0.144	0.039	2.890	**	Accepted
H6	0.224	0.044	4.158	***	Accepted
H7	−0.053	0.031	−1.166	0.244	Rejected
H8	0.201	0.046	3.689	***	Accepted
H9	0.179	0.060	3.226	***	Accepted

Note: * *p* < 0.05; ** *p* < 0.01; *** *p* < 0.001.

**Table 5 behavsci-13-00010-t005:** Path analysis.

Path	Estimate	Lower Limit	Upper Limit	Proportion of Mediation Effect
RP → AT → DE	0.264	0.169	0.358	36.4%
RP → DE → TI	0.495	0.405	0.585	17.5%

**Table 6 behavsci-13-00010-t006:** Direct effect, indirect effect, and total effect.

	Direct Effect		Indirect Effect		Total Effect	
	DE	TI	DE	TI	DE	TI
RP	0.264 ***	0.495 ***	0.151 ***	0.105 ***	0.414 ***	0.601 ***

Note: *** *p* < 0.001.

## Data Availability

The data used to support the findings of this study are available from the corresponding author upon request.

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
