# Peer review of "Research on the Influence Mechanism of Intention to Proximity Travel under the COVID-19"

_behavsci, 2022, doi:10.3390/bs13010010_

Round 1

Reviewer 1 Report

This is a very interesting paper looking at proximity travel under the COVID-19.

I compliment the authors for this well-written paper, which concerned with an important and contemporary topic.

Although in my opinion this manuscript should be published, I would suggest some changes for improving it even more, as reported below. I hope that authors will find my suggestions in a constructive way.

ABSTRACT

Results should be specified better. They are presented too succinctly and in a general manner. For example, I suggest indicating the territorial area of the research, the number of participants, and the period in which the research was carried out.

INTRODUCTION

I would like to see a more expanded discourse on why and how exploring attitude toward tourism under the COVID-19 is significant in terms of public policies. 

THEORETICAL FRAMEWORK AND LITERATURE REVIEW

I feel that authors they should improve this part. 

Since the outbreak of the pandemic, a number of interesting papers about consequences of COVID on tourism sector have been published and it is important to mention them. They pioneered this strand of studies. Among the most important, I suggest to consider:

Bakar, N. A., & Rosbi, S. (2020). Effect of Coronavirus disease (COVID-19) to tourism industry. International Journal of Advanced Engineering Research and Science, 7(4), 189-193.

Corbisiero, F., & Monaco, S. (2021). Post-pandemic tourism resilience: changes in Italians’ travel behavior and the possible responses of tourist cities. Worldwide Hospitality and Tourism Themes, 13(3), 401-417.

Gössling, S., Scott, D., & Hall, C. M. (2020). Pandemics, tourism and global change: a rapid assessment of COVID-19. Journal of sustainable tourism, 29(1), 1-20.

López Sánchez, M., Linares Gómez del Pulgar, M., & Tejedor Cabrera, A. (2021). Perspectives on proximity tourism planning in peri-urban areas. European Planning Studies, 1-18.

Monaco, S. (2021). Tourism, Safety and COVID-19: Security, Digitization and Tourist Behaviour. Routledge.

Rahman, M. K., Gazi, M. A. I., Bhuiyan, M. A., & Rahaman, M. A. (2021). Effect of Covid-19 pandemic on tourist travel risk and management perceptions. Plos one, 16(9), e0256486.

Škare, M., Soriano, D. R., & Porada-RochoÅ„, M. (2021). Impact of COVID-19 on the travel and tourism industry. Technological Forecasting and Social Change, 163, 120469.

METHODS

Authors should provide more information about recruitment strategies they used and the channels of dissemination and collection of data. They quickly talked about snowball sampling. I believe that this aspect needs to be explored instead. 

DISCUSSION

The discussion is well-done and full of interesting explanations of the results. However, at the same time, it is quite confusing, as it did not follow hypotheses. I suggest authors to rewrite the discussion on the basis of confirmation/non-confirmation of the hypotheses.

Finally, I suggest to compare the results with respect to the theoretical background and empirical research.

Author Response

Dear reviewer:

Thank you for your suggestions from which we have benefited immensely. We have revised this manuscript according to your suggestions and now believe that the article has improved in terms of logic and fluency. We have marked the revised parts in the article, using the "Track Changes" function in Microsoft Word

Suggestion 1: ABSTRACT: Results should be specified better. They are presented too succinctly and in a general manner. For example, I suggest indicating the territorial area of the research, the number of participants, and the period in which the research was carried out.

Response: Thanks for your suggestions. We have improved the abstract of the article by adding the territorial area of the research, the number of participants, and the period in which the research was carried out.

Suggestion 2: INTRODUCTION: I would like to see a more expanded discourse on why and how exploring attitude toward tourism under the COVID-19 is significant in terms of public policies.

Response: Thanks for your suggestions. By reading the relevant literature, we found that the attitude of tourism does play an important role in public policy, so we supplemented this part in the introduction.

Suggestion 3: THEORETICAL FRAMEWORK AND LITERATURE REVIEW: I feel that authors they should improve this part. Since the outbreak of the pandemic, a number of interesting papers about consequences of COVID on tourism sector have been published and it is important to mention them. They pioneered this strand of studies. Among the most important, I suggest to consider:

Response: Thanks for your suggestions. The literature you provide is a great reference for our articles. And we cited these literatures. We have refined the part of theoretical framework and literature review to make it more relevant to the topic of the article.

Suggestion 4: METHODS: Authors should provide more information about recruitment strategies they used and the channels of dissemination and collection of data. They quickly talked about snowball sampling. I believe that this aspect needs to be explored instead.

Response: Thanks for your suggestions. We've refined and supplemented our methodology section to make it more specific and detailed.

Suggestion 5: DISCUSSION: The discussion is well-done and full of interesting explanations of the results. However, at the same time, it is quite confusing, as it did not follow hypotheses. I suggest authors to rewrite the discussion on the basis of confirmation/non-confirmation of the hypotheses. Finally, I suggest to compare the results with respect to the theoretical background and empirical research.

Response: Thanks for your suggestions. We modified the discussion part according to the hypothesis of the paper, so that the discussion is closely related to the results of the article

Thank you for your time and consideration.

Yours Sincerely,

Mu Zhang

Reviewer 2 Report

A very good article, which respects the requirements of publication. Congratulations to the authors!

Author Response

Dear reviewer:

Thank you so much for your positive comments.